# Understanding challenges and enhancing the competency of healthcare providers for disability inclusive sexual and reproductive health services in rural Nepal

**Pabitra Neupane** [ORCID][1]*, **Suyasha Adhikari**[1], **Sushma Khanal**[2], **Sulochana Devkota**[3], **Manasi Sharma**[4], **Anisha Shrestha**[5], **Amit Timilsina**[6]

1 Master of Arts in Gender Studies, Tribhuwan University, Kathmandu, Bagmati Province, Nepal, 2 Central Department of Public Health, Bachelor in Public Health, Tribhuvan University, Kathmandu, Bagmati Province, Nepal, 3 Bachelor in Public Health, School of Health & Allied Sciences, Pokhara University, Kaski, Gandaki Province, Nepal, 4 Bachelor of Public Health, Manmohan Memorial Institute of Health Sciences, Tribhuvan University, Kathmandu, Bagmati Province, Nepal, 5 Centre for Karnali Rural Promotion and Society Development (CDS-PARK), Mugu, Karnali Province, Nepal, 6 Research and Community Development Centre, Kathmandu, Bagmati Province, Nepal

* neupanepavitra@gmail.com

**Data Availability Statement:** All relevant data are within the paper and its Supporting information files.

## Abstract

### Background

Women with disabilities in rural Nepal face many challenges in accessing sexual and reproductive health services including harassment and unpleasant behavior by the healthcare providers. Though National Guideline for Disability Inclusive Health Service in Nepal is in place, there are gaps in providing the sexual and reproductive health needs of women with disabilities. There has been limited research exploring the competency and capacity of health care providers on providing sexual and reproductive health services and information for women living with disabilities. Thus, this study aims to explore the competencies of rural healthcare providers in delivering disability inclusive sexual and reproductive health services for women with disabilities.

### Methods

The study used qualitative research methodology using thematic research design. Key informant interviews and focus group discussions were conducted using semi-structured interview guidelines to obtain information. Data collection was carried out till the data saturation was reached. Inductive coding was done using Dedoose software. The codebook was developed, sub-themes and themes were developed and presented as result of this study.

### Results

Knowledge gaps in disability management, procedural skills and biased perception towards sexual and reproductive health need of women with disabilities, were evident among healthcare providers. Additionally, an inadequate skill among health care providers to

**Funding:** The author(s) received no specific funding for this work.

**Competing interests:** The authors have declared that no competing interests exist.

communicate with women with disabilities for service provision was evident. To address these challenges and enhance the competency of the health care providers there is need for disability management trainings for them. Other important measures such as inclusion of disability and sexual and reproductive health in medical education curriculum, provision of sign language interpreter and disability inclusive information system, decision-making abilities and authority for disability inclusive infrastructure and tool is necessary.

## Conclusion

To address the existing challenges for disability inclusive health services, it is essential to strengthen the competency and agency of the healthcare providers, and improve the eco-system of health institution. For this, it is important for health institutions to be disability inclusive, improved behavior and attitude of health care providers, enhanced clinical knowledge on disability management and procedural skills of healthcare providers. Additionally, improving interpersonal communication skills and decision-making autonomy of health care providers is important for disability inclusive SRH services.

## Introduction

The equity report from the World Health Organization underpins the low level of health equity among women with disabilities and calls for urgent action [1]. As per the National Population and Housing Census 2021, 2.2 percent of the total population in Nepal experiences some form of disability, among which 45.8 percent are females [2]. The needs of women with disabilities include, but are not limited to, menstrual health management, family planning, maternal health, sexually transmitted infections (STIs) and Human Immunodeficiency Virus (HIV)/Acquired Immunodeficiency Syndrome (AIDS), reproductive tract infections, pelvic organ prolapse, sexual health, breast cancer, cervical cancer, etc. [3–7]. The article 38 of the Constitution of Nepal states that every woman have right to reproductive and safe motherhood, and the provisions of the right to health cover the issues of women with disabilities [8]. Similarly, the Right to Safe Motherhood and Reproductive Health Act, 2018 in Nepal which states regardless of any form of disability no one shall discriminate on the right to get disability inclusive SRH services including family planning, reproductive health, safe motherhood, safe abortion, emergency obstetric and newborn care [9]. Furthermore, the, Act Relating to Rights of Person with Disability, 2017 has urged Government of Nepal to make necessary provisions for the treatment in disability friendly environment and the protection of health and the reproductive rights taking into account of the special situation of the women with disabilities [10]. The National Guideline for Disability-Inclusive Health Services 2019, envisions to provide minimum services which includes mapping and profiling, wellness support, assistive devices, basic assessments, referral services, and networking with community rehabilitation centers, first line of counselling and early detection through primary level of health facilities (such as Health Posts, Primary Health Care Centers, Urban Health Clinics and Community Health unit) and addition services such as specialized medical treatment with test, treatment and therapies through Provincial hospitals [3].

Across the globe, the Sexual and Reproductive Health (SRH) need of women with disabilities (such as sexual desires, sexual health, need for information and services regarding family planning, abortion, maternal health, reproductive tract infection, sexually transmitted

infections, gender-based violence services, bodily autonomy etc.) are often neglected, unrecognized, and compromised, resulting in low access to SRH services [11–16]. In the context of Nepal, some studies suggest that inappropriate behavior, rudeness, and discrimination from health care providers are key factors contributing to the low utilization of SRH services [16–18]. A cross-sectional study conducted in six districts of Nepal found that 76% of young persons with disabilities were aware of important aspects of SRH, such as family planning, safe abortion, and STIs. However, 22% encountered challenges in talking to their health providers, highlighting a gap in service provision [16]. While a few studies have reviewed the healthcare experiences of women with disabilities, and minimal studies have been conducted to understand healthcare providers' attitudes toward disability in Nepal, knowledge and current practices regarding disability-inclusive and gender-sensitive SRH information and services have been less studied [19, 20]. Similarly, there are limited studies that aim to understand the distinct competency-based needs of healthcare providers to address the needs of women living with disabilities for SRH information and services. The World Health Organization (WHO) has defined competencies of healthcare providers as the contextual ability to integrate knowledge, skills, and attitudes reflected through their own behavior and actions [21]. The six domains of competencies as identified by WHO include people-centeredness, personal conduct, evidence-based practice, collaboration, communication, and decision-making [21]. Thus, this study aims to explore the challenges faced by the health care providers and ways to enhance the competencies among rural healthcare providers for disability-inclusive SRH services for women with disabilities using Global Competency Framework for Universal Health Coverage.

## Methods and materials

### Study design and setting

The qualitative research method was utilized using thematic analysis to understand the knowledge, attitudes, and practices in delivering SRH services to women with disabilities. Thematic analysis, guided by Braun and Clarke, was adapted to analyze the data and develop themes based on the data [22]. The study was conducted in four rural public health institutions where persons living with disability access SRHR services from Mugu and Surkhet district of Karnali Province of Nepal.

### Participants and sampling

Purposive sampling was conducted, where the criteria for participant selection were developed in consultation with the respective municipalities and health institutions. Prior approval was obtained from the health facility, and the participant criteria were discussed with the health facility in-charge to finalize the list of participants. For this study, purposive sampling was conducted to select healthcare providers (male and female) of age 18 and above and providing SRH services at the community level for at least a year and have experience in providing services to women with disabilities seeking SRH services. The participants who didn't provided consent to participate in the study were not included in the study.

### Data collection tools and procedures

The data was collected from 10th December 2023 to 15th January 2024. Written ethical approval was obtained from the Ethical Review Board of Nepal Health Research Council (Ref. Number 783) before conducting the data collection. A semi-structured interview guideline was developed by the researchers as a data collection tool and pretested among two

**Table 1. Socio-demographic details of participants.**

| S. N | I.D | Gender | Age | Positions | Years of Experiences |
|---|---|---|---|---|---|
| **IDI participants** | | | | | |
| 1. | H1 | Male | 31 | Medical officer | Five years |
| 2. | H2 | Male | 28 | Medical officer | One year |
| 3. | H3 | Male | 25 | Medical officer | Two years |
| 4. | H4 | Male | 33 | Doctor of Medicine, General practitioner | Two years |
| 5. | H5 | Male | 31 | Medical officer | Four years |
| 6. | H6 | Male | 27 | Medical officer | One and half years |
| 7. | H7 | Male | 28 | Medical officer | One year |
| 8. | H8 | Male | 30 | Medical officer | Two and half years |
| 9. | H9 | Female | 23 | Senior Nurse | One year |
| 10. | H10 | Female | 24 | Health Assistant | One month |
| 11. | H11 | Female | 30 | Auxiliary Nurse Midwife | Three and half years |
| 12. | H12 | Male | 26 | Health Assistant | One year |
| 13. | H13 | Male | 26 | Physiotherapist | Three and half years |
| 14. | H14 | Female | 28 | Auxiliary Nurse Midwife | Eight years |
| 15. | H15 | Female | 24 | Health Assistant | One year |
| 16. | H16 | Female | 28 | Staff Nurse | One and a half years |
| **FGD participants** | | | | | |
| 17 | H17 | Male | 28 | Medical officer | Two years |
| 18 | H18 | Male | 40 | Public health Inspector | Five years |
| 19 | H19 | Female | 36 | Sr. Auxiliary Nurse Midwife | Three years |
| 20 | H20 | Female | 36 | Sr. Auxiliary Health Worker | Four years |
| 21 | H21 | Female | 38 | Sr. Auxiliary Health Worker | Three years |
| 22 | H22 | Female | 21 | Lab Assistant | One year |
| 23 | H23 | Female | 39 | Lab Assistant | Two years |
| 24 | H24 | Female | 31 | Auxiliary Nurse Midwife | Three years |
| 25 | H25 | Female | 56 | Sr. Auxiliary Nurse Midwife | Six years |

participants for content and face validity. The data from the pretest have not been included in this study. Necessary changes in interviewing skills, such as probing healthcare providers through case-specific scenarios, were added to the interview guideline. Helsinki principles were followed throughout the study to maintain high level of research ethics in this study.

Key Informant Interviews (KII) and a Focus Group Discussion (FGD) were conducted among 25 healthcare providers (12 male participants and 13 female participants) using semi-structured interview guidelines. A total of 16 KIIs were conducted among the participants, followed by an FGD where a total of 9 healthcare providers participated. Participants of age range 24–54 participated in the study. The socio-demographic details of the participants have been mentioned in Table 1. The data were collected by the researchers themselves, who have extensive experience in qualitative data collection techniques. Data collection continued until data saturation was reached to fulfill the objective of the study, as defined by Fusch et al [23]. The semi-structured interview guide for KII and FGD were developed based on the six components of competency and outcome. The KII was aimed to understand knowledge, attitude and practice regarding SRH services to women with disabilities, while the FGD focused on ways to reduce barriers for disability- inclusive health service provision. Written consent was taken from participants before start of the interview and focus group discussion and on average interview time for KII was 38 minutes, while the FGD lasted for 69 minutes.

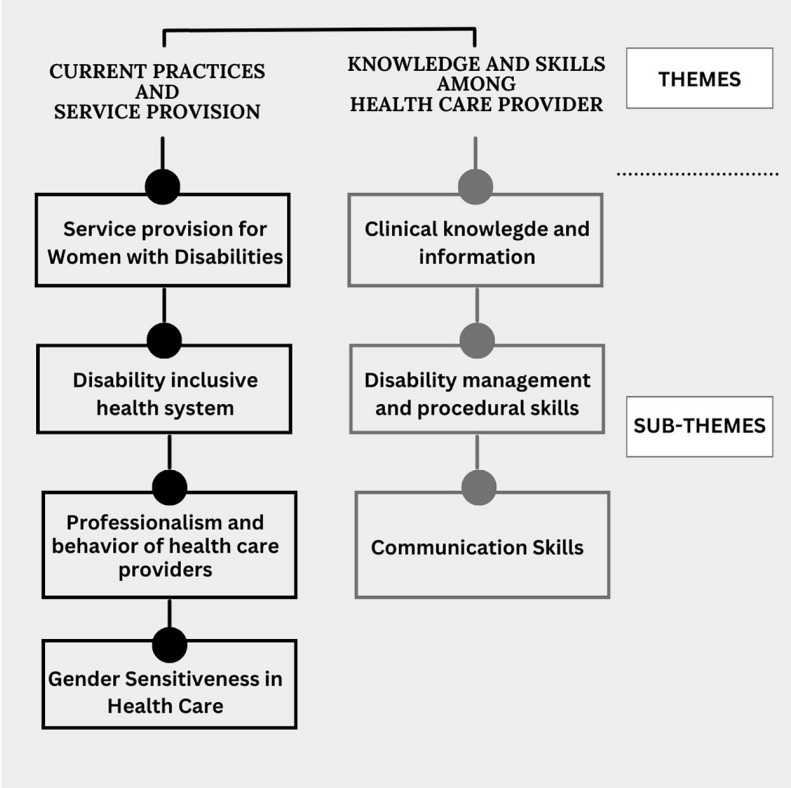

**Fig 1. The figure illustrating themes and sub-themes.**

## Data processing and analysis

The interviews were conducted in the Nepali language, translated into English, and processed for the coding of the transcripts. Inductive coding was carried out using Dedoose software, a software for managing qualitative data. A total of 68 codes were developed, and relevant themes and subthemes were identified, which have been presented as results in this study. Consultations were conducted with representatives of organizations of persons with disabilities, healthcare providers, and local government to cross-validate the findings.

# Result

A total of 68 codes and two themes and six sub-themes: Current Practices and Service Provision (sub-theme: service provision for women with disabilities, disability inclusive health system, professionalism and behavior of health care providers) and Knowledge and Skills Among Health Care Providers (sub-theme: clinical knowledge and information, disability management and procedural skills, interpersonal relationship and communication) were generated as stated in Fig 1: The figure illustrating themes and sub-themes.

## Current practices and service provision for disability-inclusive health services

The existing practices and service provision for disability-inclusive health services were analyzed by considering the ongoing initiatives, procedures, and services provided by healthcare

professionals when delivering care to women with disabilities, particularly in the context of their SRH.

**Service provision for women with disabilities.** Few healthcare providers mentioned prioritizing women with disabilities in various healthcare settings, including in queues, emergency wards, outpatient departments, laboratories, and malnutrition units, to ensure they receive timely and appropriate services. Similarly, free medical services and nutritional support, such as RUTF (ready-to-use therapeutic food), along with referral mechanisms, have also been provided as specialized care. A 31 years old male, Medical Officer (H5) stated:

*Patients/clients with disabilities are prioritized compared to general patients. Other must wait in the queue, but patients/clients with disabilities don't have to wait, we treat them first. We prioritize patients with disabilities for treatment, investigation, and medication.*

One of the healthcare providers mentioned providing healthcare services based on the specific needs of women with disabilities such as providing online reports to minimize the physical burden on women with disabilities and arranging mobility devices like wheelchairs and crutches based on the client/patients' needs, demonstrating a client/patient-centric approach. A 24 years old female, Health Assistant (H15) shared:

*We provide services according to the case and need. We offer online reports to minimize the need for clients/patients with disabilities to travel. Only in cases of emergency clients/patients with disabilities are brought to the hospital by carrying on a stretcher.*

Few participants of this study provide information and education on Sexual and Reproductive Health and Rights (SRHR) to women with disabilities, demonstrating a commitment to equity and inclusivity in healthcare provision, regardless of an individual's disability. One healthcare provider discussed her experience in counseling women with disabilities during Antenatal Care (ANC) visits. She adapted her counseling methods, recognizing the need for personalized approaches based on the client/patient's disability. A 23 years old female Senior Nurse (H9) stated:

*If women with disabilities are not educated or aware of SRHR, then we provide them with information on this. Women with disabilities have rights to family planning devices, so I don't deprive them of such services despite their disabilities. I ensure that women with disabilities receive services for their SRH through counseling and facilitation.*

**Disability-inclusive health system.** Few healthcare providers were aware about the need for disability inclusive SRH services and suggested that advocating for government policies and regulations that prioritize the needs of persons with disabilities should be encouraged to ensure healthcare facilities are more accessible and inclusive for persons with disabilities. Collaboration among government agencies, NGOs, and INGOs should be ensured to strengthen the health system for increased investment in enhancing the competency of health care providers and creating an enabling environment for competent healthcare providers to deliver quality SRH services among women with disabilities. A 30 years old male, Medical Officer (H8) shared:

*To make sure persons with disabilities are safe, Nepal government should pay attention, and development organizations should do their jobs well. They should run programs and give*

*training to health care providers to enhance their competency to provide disability inclusive services.*

Likewise, the majority of healthcare providers stated that the existing superstitions and deeply rooted harmful cultural practices restrict women with disabilities from seeking SRH services. One healthcare provider suggests the importance of disseminating SRH information among women with disabilities to debunk myths and harmful practices at the grassroots level, and for this deploying trained healthcare providers on disability-inclusive services at health posts is crucial. A 24 years old female, Health Assistant (H10) stated:

*Superstitions and the deeply rooted harmful culture of rural areas play a significant role in shaping the health-seeking behavior of people, including women with disabilities. Therefore, awareness programs in grassroot level should be conducted to instill trust in health service provision among women with disabilities and for this it's crucial to train health care providers of health posts.*

Majority of the healthcare facilities in rural Nepal encounter difficulties in addressing the needs of persons with disabilities and ensuring that service provision is people-centered. These challenges arise from the inadequate infrastructure in health facilities (health posts and hospitals) which lacks necessary equipment, including Computed Tomography (CT) scans, Magnetic resonance imaging (MRIs), and disability-specialized diagnostic tools in hospitals. Thus, government policies ensuring basic medical services, these facilities struggle to acquire disability-specialized machinery and tools due to limited budgets. A 31-year-old male, Medical officer (H5) shared that:

*There is the government policy which has mentioned that the infrastructure of health services must be made accessible to persons with disabilities, but we still lack the necessary resources and equipment to provide all the mentioned services to them.*

Furthermore, healthcare providers, including Medical Superintendents, lack decision-making power to develop disability-inclusive infrastructure or purchase tools and equipment independently due to budget constraints, unless supported by development partners. A 30 years old male, Medical Officer (H8) states:

*Government does not provide wheelchairs; and other materials and we don't have decision making power and budget to buy ourselves, we get it when some organizations support us to buy.*

The participants who have taken the disability management and sensitization trainings from the provincial government has highlighted increased competence and leadership to develop their health facility disability friendly infrastructures (tactile footpath, ramp, accessible railings, accessible toilets, obstacles in entrances) and disability inclusive information, Education and Communications (IEC) materials (in braille, large print, audio-visual devices, pictorial posters) in support of local government and development partners. A 40 years old male, Public Health Inspector (H18) stated:

*After attending the Disability Management and Sensitization Training organized by one of the development partners, I gained a deeper understanding that accessibility extends beyond physical structures like ramps. It encompasses ensuring access to information and positive*

*behaviors towards persons with disabilities. After the training, in support of Community-Based Organizations, I have initiated the installment of braille printed signage in the hospital, and integrated disability related data into the Health Management Information System.*

**Professionalism and behavior of health care providers.** The attitude of healthcare workers refers to the beliefs, perceptions, and behaviors of healthcare professionals when providing care to women with disabilities in the context of SRH. A range of attitudes and behaviors were observed among healthcare providers. Few demonstrated a positive attitude towards clients/patients with disabilities while providing services, and they also suggested essential measures to address inappropriate behavior towards such clients/patients. Some expressed a commitment to treating persons with disabilities with respect and care, while others were less attentive or empathetic. A 28 years old male, Medical Officer (H2) stated:

*It is very difficult to counsel clients/patients with disabilities compared to abled clients/patients. It's difficult to make them understand about the prescribed dose of medicine.*

Some of the participants also shared that sometimes, healthcare providers may also face frustration as they might feel overwhelmed by heavy workloads resulting to impolite interactions with clients/patients with disabilities. A 28 years old Auxiliary Nurse Midwife (H14) stated:

*Clients/patients with disabilities may not grasp information as quickly as other clients/patients, which could contribute to frustrations among healthcare provider due to heavy workload.*

The participants of the FGD expressed their opinions on whether there are differences in healthcare needs between women with disabilities and those without. Some providers acknowledged variations, while others stated that there are no significant disparities. Despite the differences in perspectives, the majority of the healthcare providers in the FGD emphasized the importance of providing equal services to both women with disabilities and those without. They stated that all women should have access to the same standard of care. 36 years old female, Sr. Auxiliary Nurse Midwife (H19) stated

*I have always provided equal services to women with and without disabilities. Yes, there is still a misconception regarding the need of SRH services to women with disabilities as among us also, we sometimes doubt if they actually need it or not. However, while providing the services, I have always practiced of providing same services.*

One of the healthcare providers stated that the attitude and behavior of every individual healthcare provider cannot be generalized based on some examples. Likewise, some of the participants also mentioned the correlation of time constraint with the behavior of health care providers, indicating that due to time constraints and complicated communication with client/patients with disabilities, they sometimes happen to behave rudely. A 26 years old male, Physiotherapist (H13) shared:

*It's not appropriate to generalize about the behavior of all healthcare providers based on a few examples. Healthcare professionals often face time constraints when treating large number of*

*clients/patients, and when communication between doctors and clients/patients with disabilities becomes complex, doctors are sometimes forced to behave rudely.*

Some of the healthcare providers were aware of the importance of better communication and creating a disability-inclusive environment while providing services to women with disabilities. One healthcare provider highlighted the importance of guiding staff to assist clients/patients after they leave the Outpatient Department (OPD). He mentioned that it's challenging to track the movement of clients/patients after they leave the OPD and the emergency area. A participant also shared that there is misunderstanding among clients/patients as they perceive every health office staff as a 'doctor', which results in false reporting about the bad attitude of medical doctors towards the clients/patients. A 28 years old female, Staff Nurse (H16) stated:

*In this hospital, clients/patients with disabilities often mistake all the staff members for doctors and they might face rude behavior from those staffs, which is primarily a result of their misunderstanding.*

It was found that there was a gender bias in the level of care provided to persons with disabilities by healthcare providers. Limited effort is made to communicate effectively with women with disabilities and to understand the needs of women with disabilities while assessing SRH services. Some health care providers even sent women back to bring a companion, despite them having to travel long distances for health services. A 31 years old female, Auxiliary Nurse Midwife (H24) stated:

*Once I had an experience of counseling a woman with a disability for ANC who couldn't respond quickly and couldn't speak, then I asked to come up with her husband next time.*

## Knowledge and skills among the health care providers

The knowledge of healthcare providers towards women with disabilities seeking sexual reproductive health services was analyzed based on the providers' understanding and awareness of the specific SRH needs and challenges faced by women with disabilities while seeking SRH services.

**Clinical knowledge and information.** Some of the healthcare providers were aware of the categorization of persons with disabilities, while few of them were not fully aware of the categories of disability. A 27 years old male, Medical Officer (H6) stated:

*Honestly, I don't know have knowledge regarding categories of disabilities and their issues. I have limited knowledge since I haven't got any disability-oriented training related to this. I know we should be kind toward them while providing the services.*

Majority of the healthcare providers suggested that women with disabilities have distinct needs and require additional attention and care for SRH services compared to women without disabilities. A 25 years old male, medical officer (H3) stated:

*Women with disabilities belong to a vulnerable group. Their health needs differ as their body formation is also different from that of other women. We have to provide more time and concern while dealing with these women.*

Most of the time, healthcare providers rely on observation of physical deformities or external injuries to identify the patient's issues without consulting them. Healthcare providers seek support from female nurses to provide services and support for women with disabilities. A 31 years old male, Medical Officer (H1) stated:

*If a patient is unable to speak and hear, then we try to communicate through sign language as far as possible. In case we can't communicate through sign language, we will identify the problem of the patient through observation, such as observing physical deformities or external injuries.*

Some of the healthcare providers were partially aware about the provision of free health services provided to persons with disabilities and few of them were unaware about it as they lacked orientation about the free health services for persons with disabilities. A 26 years old male, Health Assistant (H12) stated:

*We are not provided orientation about provision of free services to persons with disabilities. Because of that, staff members lack awareness of the available free services for persons with disabilities. As a result, some clients/patients with disabilities might have missed out on accessing the free services they are entitled to.*

**Disability management and procedural skills.**    Majority of the healthcare providers lacked procedural knowledge and skills in treating clients/patients with disabilities as they expressed that they had not received any specific training on how to treat clients/patients with disabilities. They strongly acknowledged the need for such training, recognizing that a lack of training might be responsible for the current situation where all clients/patients are treated similarly, without considering their specific conditions. Some participants reflected on the need to provide training on sign language or availability of sign language interpreter in health institutions for effective communication. A 30 years old female, Auxiliary Nurse Midwife (H11) stated:

*I haven't heard and received any training on how to treat patient/client with disabilities. We treat clients/patients with our understanding and knowledge. I believe that every health professionals need this training to provide better services to client/patients with disabilities as population of persons with disabilities is high in hilly region.*

Some of healthcare workers received verbal consent and shared practicing privacy with women with physical disabilities or with their closest relatives. However, with literate hard-of-hearing clients/patients, it was found to practice written consent to create an enabling environment for them. Some healthcare providers also reported not taking any consent from women with physical disabilities except for caesarean section surgery, and they seem to use non-verbal language as the healthcare providers do not have any sign language training to communicate with the client/patient. A 38 years old female, Auxiliary Health Worker (H21) stated:

*For clients/patients with visual impairment, we take verbal consent. It is challenging when dealing with hearing-impaired clients/patients as we don't know sign language. However, we make efforts to communicate with them through written form, particularly for those who are literate.*

Only one healthcare provider mentioned about receiving disability sensitization training, but the majority of the healthcare providers stated that they haven't received any such training due to which they face difficulties while dealing with clients/patients with disabilities in general, and women with disabilities in particular, compromising the quality of health services provided to women with disabilities. A 33 years old male Doctor of Medicine, General practitioner (H4) stated:

> *Specific training related to disability should be provided. As far as I know, no doctors or nurses have received such specific training on disability and are also unaware of it. So, if there is any such training, please inform us, and we will take the training.*

**Communication skills.**   Some healthcare providers imparted information and education to clients/patients with hearing impairments using translators and engaging close relatives to facilitate communication when translators were not available. A 36 years old, Sr. Auxiliary Health Worker (H20) stated:

> *There are students with disabilities in a school here who have hearing impairments. They come for checkups and they come with a translator. So, it becomes easier to deal with them. For some clients/patients who don't have a translator, we ask the closest relative or patient's party to communicate.*

Some of the participants highlighted the need for sign language interpreter, audio-visual IEC materials, SRH information printed in Braille for easy, effective and adequate information regarding SRH to women with disabilities. A 28 years old male, Medical Officer (H17) stated:

> *I encountered a 35-year-old woman who was deaf and mute, and she had a large belly. I faced communication barrier as I didn't know sign language, I was unsure if she had stomach issues or pregnancy. If there was a sign language interpreter, I could understand her issues.*

## Discussion

The findings of this study highlight limited clinical knowledge and disability management skills, limited skills for effective communication with clients/patients with disabilities, low gender sensitivity, and lack of evidence-based and practice-based learning mechanisms within the healthcare system. The National Guideline for Disability Inclusive Health Services 2019 ensures that health care providers should ensure the integration of SRHR needs and rights of persons with disabilities in existing program and service delivery at each service provision level [3]. Furthermore, also highlights the capacity of health care providers to recognize and respect the SRH need of people with disabilities, raise awareness regarding SRHR and availability of SRH services among people with disabilities. Contrasting to the expected knowledge and skillset for the disability inclusive SRH services as envisioned by The National Guideline for Disability Inclusive Health Services; this study suggests that health care providers have limited knowledge and information regarding the types of disability, the consenting process, disability-inclusive healthcare, and the SRH needs of women with disabilities. The findings of this study are similar to multiple studies that have shown a low level of knowledge and understanding regarding disabilities among healthcare providers [20, 24]. The limited knowledge and understanding regarding disability among healthcare providers could be an outcome of the

limited inclusion of a comprehensive disability-related curriculum in the medical education system of Nepal. The education curriculum includes information and knowledge regarding clinical components of disability and rehabilitation in the medical curriculum of various disciplines (MBBS, Health Assistant, Proficient Certificate Level Nursing) [25, 26]. However, the findings of this study suggests that the current contents in medical education have not equipped health care providers with the necessary knowledge (such as categorization of disabilities, health service provision for persons with disability, health policy and program related to person with disability, disability and SRH) and skills (such as communicating with women with disability, ethics and consent, referral mechanism, management and decision making) to strengthen competency of health care providers to provide minimal services including SRH services to women with disabilities as suggested by the National Guideline for Disability Inclusive Health Services 2019. These findings from this study suggest clear gap between scope of education and required knowledge and skills while practicing at field to cater the SRH need of women with disabilities which can be addressed through inclusion of disability inclusive content in curriculum, short courses and field experiences, disability inclusive management training and refresher training for health care providers. One consistent demand from healthcare providers in this study has been for disability management training and orientation. Multiple studies conducted in various countries have highlighted the need for training healthcare providers on disability health management [19, 26, 27].

Various studies conducted in Nepal from the client's perspective provide evidence of harassment, rude behavior, and unpleasant experiences of women with disabilities, suggesting unprofessionalism and gender insensitivity among health care providers [18, 28, 29]. In response to previous research findings, medical doctors in particular in this study have reported being misunderstood with 'other healthcare providers' from health institutions and thus labeled often as 'rude and unprofessional' by the client. Thus, this study recommends study including both clients (persons with disabilities) and healthcare providers from the same institutions to address the prevalent perception of harassment and unpleasant behavior between providers and clients. Furthermore, the report of 'misunderstanding with other healthcare providers' also reflect the need to orient, train, capacitate and empower entire health workforce including medical doctors, nurses, paramedics, mid-level health workers, and other support team within the health institution to destigmatize the healthcare providers and for the provision of disability inclusive health service.

This study suggests inadequate provision and investment in disability-friendly infrastructures (such as ramp, railings, tactical path, signages, disability friendly IEC materials), equipment, commodities and tools (such as MRI and CT Scan in provincial hospital and Ultra Sound Sonography, fetal doppler, Pap-Smear test kit, Silicon Ring Pessary in primary hospitals). Furthermore the limited disability management training for healthcare providers compromises knowledge, skills and attitude of health care providers which has eluded the right of women with disabilities for quality SRH services, and to enhance the competency of healthcare providers as stated in the National Health Policy 2019 and The National Guidelines of Disability Health Services 2019 [3, 30]. The National Guidelines of Disability Health Services 2019, acknowledges the responsibility and role of local and provincial government to implement phased plan for disability inclusive health service to review and periodically assess the capacity of health care providers, map and develop profile of health institution to ensure disability inclusiveness, strengthen the coordination and collaboration for support [3]. The effective implemented of the guideline could help address the existing barriers of competency of health care providers including other service providers at institutions, allocating adequate resources for disability inclusive infrastructure and equipment, access to IEC materials, transferring skills regarding sign language, and decentralization of services and decision making. However,

in Nepal, there has been no mechanism yet to include data regarding women with disabilities receiving health services, including SRH services, in the Health Management Information System which has compromised prioritization of disability inclusive SRH service provision and evidence-based decision-making in the health system to implement existing policy, program and plans [31, 32].

The Government of Nepal has highlighted provision of disability inclusive training during induction and in-service training provided through National Health Training Centers but only few have recently initiated efforts to provide such training to health care providers using the Disability Management and Rehabilitation Training Manual [31, 32]. Similarly, few provincial health directorates in Nepal have recently conducted training on Rehabilitation Clinical Protocol, Rehabilitation Service Standard Operating Procedures, Assistive Product List, and National Standard Assistive Technology [31, 32]. The Disability Management and Rehabilitation Training Manual includes topics such as types of disabilities, dealing with clients/patients with disabilities, orientation regarding national social and legal scenarios, and provisions for disability. These topics aim to equip healthcare providers with the necessary knowledge and skills to provide disability-inclusive SRHR services [32]. The positive outcomes of training, such as increased knowledge, understanding, and commitment to disability-inclusive service among healthcare providers, suggest the need to scale up of such disability inclusive training programs across Nepal [31]. These programs should target strengthening the competency of healthcare providers in delivering disability inclusive SRH services.

## Strengths and limitations of the study

The findings of the study such as self-perceived competency regarding disability and SRHR, existing barriers to provide disability inclusive SRH services for women with disabilities and understanding the need of health care providers are unique findings for Nepal. The study adequately reflects and discusses the gap in current policy, act, and guideline which is another important contribution. These findings could be helpful for policy makers, local and provincial government decision makers, program implementers and health care providers to bridge the existing gap and ensure enabling environment for disability inclusive SRH services.

The study reflects the experience and opinion of health care providers from Mugu and Surkhet districts of Karnali province. Thus, the findings of the study may not necessarily reflect the views of health care providers across the country. This study does not intend to analyze the disability related content in the medical curriculum and its application in the field in depth but reports the gap as stated by the participants of the study. Thus, a national level mixed-method study is recommended to analyze the content of existing curriculum, map the skills and competencies among health care providers including health managers to provide disability inclusive health services, and conduct disability inclusive infrastructure audit to assess, understand and strengthen disability inclusive health system in Nepal.

## Conclusion

The health care providers have inadequate knowledge, and skills regarding disability-inclusive care leading to poor SRH services, discrimination, poor service accessibility, and ineffective communication barriers while providing services to women with disabilities. To address these issues, it is essential to strengthen the competency of healthcare providers and improve the health institution ecosystem by equipping health care providers (and other healthcare providers) with disabilities management and procedural skills, provision of sign language interpreter, disability inclusive information tools, fostering a respectful healthcare culture, evidence based

decision making abilities and authority to ensure disability inclusive infrastructure and tools at health institutions for equitable SRH services among women with disabilities.

## Supporting information

**S1 File.**
(PDF)

## Acknowledgments

The research team would like to thank health facilities, health facility in-charge and all the participants for their valuable support and participation in this study. We would also like to thank CORE Group for their support in this study.

## Author Contributions

**Conceptualization:** Pabitra Neupane, Amit Timilsina.

**Data curation:** Pabitra Neupane, Suyasha Adhikari, Sushma Khanal, Sulochana Devkota, Manasi Sharma, Anisha Shrestha.

**Formal analysis:** Pabitra Neupane, Suyasha Adhikari, Anisha Shrestha.

**Investigation:** Pabitra Neupane, Suyasha Adhikari, Sushma Khanal, Sulochana Devkota, Manasi Sharma, Anisha Shrestha.

**Methodology:** Pabitra Neupane, Amit Timilsina.

**Project administration:** Pabitra Neupane.

**Software:** Suyasha Adhikari, Sushma Khanal, Sulochana Devkota.

**Supervision:** Amit Timilsina.

**Validation:** Amit Timilsina.

**Visualization:** Pabitra Neupane.

**Writing – original draft:** Pabitra Neupane, Suyasha Adhikari, Sushma Khanal, Sulochana Devkota, Manasi Sharma, Anisha Shrestha, Amit Timilsina.

**Writing – review & editing:** Pabitra Neupane, Amit Timilsina.

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
