## [Decision Letter · Decision Letter 0]

7 Jun 2024

PONE-D-24-16726Understanding challenges and enhancing the competency of healthcare providers for disability inclusive and gender sensitive Sexual and Reproductive Health Services in rural NepalPLOS ONE

Dear Dr.  Neupane,

Thank you for submitting your manuscript to PLOS ONE. After careful consideration, we feel that it has merit but does not fully meet PLOS ONE’s publication criteria as it currently stands. Although the points they pointed out seems minor, reviewers raised important points to be addressed as a major concern to improve your manuscript and adhere  the journal's standard. Therefore, we invite you to submit a revised version of the manuscript that addresses the points raised during the review process.

We look forward to receiving your revised manuscript.

Kind regards,

Yitagesu Habtu Aweke, Ph.D

Academic Editor

PLOS ONE

4. We note you have included a table to which you do not refer in the text of your manuscript. Please ensure that you refer to Table 1 in your text; if accepted, production will need this reference to link the reader to the Table.

Reviewers' comments:

Reviewer's Responses to Questions

**Comments to the Author**

1. Is the manuscript technically sound, and do the data support the conclusions?

Reviewer #1: Partly

Reviewer #2: Yes

2. Has the statistical analysis been performed appropriately and rigorously? 

Reviewer #1: N/A

Reviewer #2: N/A

3. Have the authors made all data underlying the findings in their manuscript fully available?

Reviewer #1: No

Reviewer #2: Yes

4. Is the manuscript presented in an intelligible fashion and written in standard English?

Reviewer #1: Yes

Reviewer #2: Yes

5. Review Comments to the Author

Reviewer #1: 1) It would be useful for the reader to understand what the national guideline for disability-unclusive services in Nepal says about providing SRH services to women with disabilities. It is unclear whether there are already provisions but the study is exploring the challenges to implementing these, or whether the guidelines are incomplete/insufficient and the challenges being explored in the study go beyond these.

In the same vein, it would be important to clarify whether the authors have a specific understanding of what healthcare provider competency entails in this case, and whether they know what the services provided to women with disabilities ought to be.

The sub-topic on gender sensitiveness should be developed further to ensure it is well argued and supported. As it stands now, it is unclear how the conclusion that there is a gender bias in care is arrived at.

When describing the results on healthcare provider knowledge it would be important for the reader to know what they are expected to know. Overall, concluding that healthcare providers have limited knowledge and skills needs to be sustained by contrasting the findings with what they should know/the expected competencies they should hold. For instance, in the discussion the following claim is made: "Though there is inclusion of some clinical components of disability in the curriculum under the general practice subject, the contents are not adequate to understand and cater to the SRH needs of women or to provide disability inclusive SRH services to women". Although the authors may know what the adequate contents should be, the reader may not and it is important to make this explicit so that the reader can follow and draw their own conclusions.

Many of these issues could be addressed by providing some context on healthcare for people living with disabilities drawn from documentary analysis (policy, guidelines).

4) The text is intelligible and overall clear, although minor grammar mistakes can be found throughout and should be addressed - for example, the use of the term 'adopted' when it is clear what is meant is 'adapted'. The formulation of sentences should be revised to improve clarity. The manuscript should be revised and edited accordingly.

Reviewer #2: The authors should clarify the following points in the manuscript:

1. In Study design and setting: the authors should clarify how many municipalities and health institutions of the districts were covered by the study?

2. In Line no. 125: For which level of health professional did the authors intend to refer by the term "health in-charge"?

3. In Line no. 224: The authors mentioned the finding that there is lack of equipments including CT and MRI for the disability services. Authors need to clarify the need of these highly valued equipments for SRH in rural setting for disability.

6. PLOS authors have the option to publish the peer review history of their article (what does this mean?). If published, this will include your full peer review and any attached files.

Reviewer #1: **Yes: **Yara Alonso Menendez

Reviewer #2: **Yes: **Basant Adhikari

---

## [Author Response · Author response to Decision Letter 0]

17 Jul 2024

17th July 2024

Yitagesu Habtu Aweke, Ph.D

RE: Revision of Manuscript PONE-D-24-16726

Dear Editor,

We would like to thank you and the reviewers for your time and providing constructive comments on our manuscript entitled “Understanding challenges and enhancing the competency of healthcare providers for disability inclusive Sexual and Reproductive Health Services in rural Nepal.”

A revised version (clean and track changes) reflecting the point-by-point response to the reviewers’ comments have been submitted for your consideration. We believe that we have addressed all the comments and that the manuscript has significantly improved.

Should you require any additional information, please do not hesitate to let me know. Thank you very much for your time and kind considerations. 

Your sincerely,

Pabitra Neupane

RESPONSE TO THE REVIEWERS

Journal Requirements:

RESPONSE: All necessary adjustments have been made. 

2. We note that your Data Availability Statement is currently as follows: All relevant data are within the manuscript and its supporting informations.

RESPONSE: The codebook produced as the outcome of the data analysis, the interview guideline for the KII and FGDs and the informed consent forms have been available for this study. As participants have not consented for the available of full interview transcripts during this study, it has not been publicly available to respect participants anonymity and consent.

RESPONSE: This section has been updated in Methodology and it reads as: the data was collected from 10th December 2023 to 15th January 2024. Written ethical approval was obtained from the Ethical Review Board of Nepal Health Research Council (Ref. Number 783) before conducting the data collection. The research follows standard ethical principles guided by the Helsinki Principles. The autonomy and confidentiality of participants has been maintained throughout the project period. All the participants were oriented about the study and its importance. Prior to data collection, written consent was taken from the participants after orientation, verbal consent was taken before starting the interviews and their right to withdraw from the research was reiterated. The participants were assured of anonymity and confidentiality before, during and after the study. 

4. We note you have included a table to which you do not refer in the text of your manuscript. Please ensure that you refer to Table 1 in your text; if accepted, production will need this reference to link the reader to the Table.

RESPONSE: This comment has been addressed in line number 167 

RESPONSE: All the supporting files has been made available in the manuscript so this is not applicable for this study. 

Reviewer’s Comment

Reviewer #1: 

1) It would be useful for the reader to understand what the national guideline for disability-inclusive services in Nepal says about providing SRH services to women with disabilities. It is unclear whether there are already provisions, but the study is exploring the challenges to implementing these, or whether the guidelines are incomplete/insufficient and the challenges being explored in the study go beyond these.

Response: Thank you so much for your comment. The additional policy provision and mechanism has been adequately mentioned in the introduction section from line numbers 76-93. This comment has been addressed completely.

2) In the same vein, it would be important to clarify whether the authors have a specific understanding of what healthcare provider competency entails in this case, and whether they know what the services provided to women with disabilities ought to be.

Response: Thank you so much for your feedback. The specific understanding of what healthcare providers competency is expected has been discussed in discussion section from line number 440-447. This comment has been addressed completely.

3) The sub-topic on gender sensitiveness should be developed further to ensure it is well argued and supported. As it stands now, it is unclear how the conclusion that there is a gender bias in care is arrived at.

Response: We discussed on this as a team, and we agree that this study does not have enough evidences to establish lack of gender sensitiveness apart from one or two excerpts. Thus, we have removed 'gender sensitiveness' from the topic and sub-theme and removed 'gender bias' from conclusion to avoid any misleading information and discussion. It is discussed under discrimination but not as a separate sub-section. This comment has been addressed.

4) When describing the results on healthcare provider knowledge it would be important for the reader to know what they are expected to know. Overall, concluding that healthcare providers have limited knowledge and skills needs to be sustained by contrasting the findings with what they should know/the expected competencies they should hold. For instance, in the discussion the following claim is made: "Though there is inclusion of some clinical components of disability in the curriculum under the general practice subject, the contents are not adequate to understand and cater to the SRH needs of women or to provide disability inclusive SRH services to women". Although the authors may know what the adequate contents should be, the reader may not and it is important to make this explicit so that the reader can follow and draw their own conclusions. Many of these issues could be addressed by providing some context on healthcare for people living with disabilities drawn from documentary analysis (policy, guidelines).

Response: Thank you so much for your comment. The additional policy provision and mechanism has been adequately mentioned in the introduction and discussion section.

4) The text is intelligible and overall clear, although minor grammar mistakes can be found throughout and should be addressed - for example, the use of the term 'adopted' when it is clear what is meant is 'adapted'. The formulation of sentences should be revised to improve clarity. The manuscript should be revised and edited accordingly.

Response: The manuscript has been thoroughly revised and checked for any language and grammatical error. This comment has been addressed.

Reviewer #2:

 The authors should clarify the following points in the manuscript:

1. In Study design and setting: the authors should clarify how many municipalities and health institutions of the districts were covered by the study?

Response: The number of rural/municipalities and health facilities has been mentioned. However, the name of municipalities and health institutions has not been mentioned to ensure anonymity and confidentiality of health institutions and facilities. The socio-demographic information is in detail and naming the municipalities and health institutions could lead to participant identification. 

2. In Line no. 125: For which level of health professional did the authors intend to refer by the term "health in-charge"?

Response: The term "health in-charge" has been corrected to "health facility in-charge" for clarity. The health facility in-charge are usually medical doctors, H.A, or senior AHW. 

3. In Line no. 224: The authors mentioned the finding that there is a lack of equipments including CT and MRI for the disability services. Authors need to clarify the need of these highly valued equipment for SRH in rural setting for disability.

Response: The health facility under the study are health posts, Primary Health Care Centers and Hospitals at district/province level and the need for equipment, commodity and tool varies. The CT and MRI are required for abdominal or pelvic screening/assessment at district level while the need of health post include USG, fetal doppler, Pap-Smear test kit, Silicon Ring Pessary. This has been clarified in the discussion section.

---

## [Decision Letter · Decision Letter 1]

24 Sep 2024

PONE-D-24-16726R1Understanding challenges and enhancing the competency of healthcare providers for disability inclusive Sexual and Reproductive Health Services in rural NepalPLOS ONE

Dear Dr. Neupane,

Thank you for submitting your manuscript to PLOS ONE.  After careful consideration, we feel that it has merit but does not fully meet PLOS ONE’s publication criteria as it currently stands. Reviewer #4 has still concerns.  Therefore, we invite you to submit a revised version of the manuscript that addresses the points raised during the review process. Please submit your revised manuscript by Nov 08 2024 11:59PM. If you will need more time than this to complete your revisions, please reply to this message or contact the journal office at plosone@plos.org. Please include the following items when submitting your revised manuscript:A rebuttal letter that responds to each point raised by the academic editor and reviewer(s). You should upload this letter as a separate file labeled 'Response to Reviewers'.A marked-up copy of your manuscript that highlights changes made to the original version. You should upload this as a separate file labeled 'Revised Manuscript with Track Changes'.An unmarked version of your revised paper without tracked changes. You should upload this as a separate file labeled 'Manuscript'.If applicable, we recommend that you deposit your laboratory protocols in protocols.io to enhance the reproducibility of your results. Protocols.io assigns your protocol its own identifier (DOI) so that it can be cited independently in the future. For instructions see: https://journals.plos.org/plosone/s/submission-guidelines#loc-laboratory-protocols. Additionally, PLOS ONE offers an option for publishing peer-reviewed Lab Protocol articles, which describe protocols hosted on protocols.io. Read more information on sharing protocols at https://plos.org/protocols?utm_medium=editorial-email&utm_source=authorletters&utm_campaign=protocols.

We look forward to receiving your revised manuscript.

Kind regards,

Yitagesu Habtu Aweke, Ph.D

Academic Editor

PLOS ONE

Journal Requirements:

Reviewers' comments:

Reviewer's Responses to Questions

**Comments to the Author**

1. If the authors have adequately addressed your comments raised in a previous round of review and you feel that this manuscript is now acceptable for publication, you may indicate that here to bypass the “Comments to the Author” section, enter your conflict of interest statement in the “Confidential to Editor” section, and submit your "Accept" recommendation.

Reviewer #3: All comments have been addressed

Reviewer #4: (No Response)

2. Is the manuscript technically sound, and do the data support the conclusions?

Reviewer #3: Yes

Reviewer #4: (No Response)

3. Has the statistical analysis been performed appropriately and rigorously? 

Reviewer #3: Yes

Reviewer #4: (No Response)

4. Have the authors made all data underlying the findings in their manuscript fully available?

Reviewer #3: Yes

Reviewer #4: (No Response)

5. Is the manuscript presented in an intelligible fashion and written in standard English?

Reviewer #3: Yes

Reviewer #4: (No Response)

6. Review Comments to the Author

Reviewer #3: The authors have successfully addressed the feedback provided by the previous reviewers, demonstrating commendable engagement with critical suggestions. Upon reviewing this revised submission, I recommend it for publication. This study is particularly notable for its focus on a critically underexplored area—disability-inclusive sexual and reproductive health services in rural Nepal. The authors' examination of healthcare providers' competencies in delivering these services is highly relevant, especially given the persisting barriers faced by women with disabilities, including knowledge gaps in disability management, biased perceptions, and inadequate communication skills.

The use of a qualitative, thematic research design, supported by key informant interviews and focus group discussions, has allowed for an in-depth exploration of these complex issues. The findings, which highlight significant gaps in procedural skills and the need for disability-inclusive training and infrastructure, provide valuable insights for improving health systems in underserved regions.

Considering that the authors are university students, the depth and rigor of their work are commendable. They have contributed meaningfully to an area of research that has clear implications for both policy and practice in the context of rural health services in Nepal. Therefore, this manuscript contributes to the field and should be published.

Reviewer #4: Abstract

• Result (Line 38-45): This sentence is long and hard to follow. It's important to clarify this for the reader to easily grasp the meaning.

• Conclusion (Line 46-50): Instead of writing one long sentence (which spans four lines), consider breaking it into simpler sentences to improve clarity.

Introduction

• Line 90: There are two periods (dots).

Method (Lines 145-150):

Participants and Sampling:

• The criteria for inclusion/exclusion need more details. Do you include both males and females? What is the age range? What languages do participants speak (Nepali or English)?

Results

• Line 176: Double-check the spacing between words.

• Lines 208-210: This sentence is somewhat unclear. Are healthcare providers aware of advocacy?

Discussion:

• “Professionalism and Behavior of Health Care Providers”: This section presents clear points and is easier to follow compared to the previous section, “Disability-Inclusive Health System,” which I have already commented on for revision.

Line 428: There is an extra period (dot).

7. PLOS authors have the option to publish the peer review history of their article (what does this mean?). If published, this will include your full peer review and any attached files.

Reviewer #3: **Yes: **Dr. Shamim Ahmed

Reviewer #4: No

---

## [Author Response · Author response to Decision Letter 1]

27 Sep 2024

27th September 2024

Yitagesu Habtu Aweke, Ph.D

RE: Revision of Manuscript PONE-D-24-16726R2

Dear Editor,

We would like to thank you and the reviewers for your time and providing constructive comments on our manuscript entitled “Understanding challenges and enhancing the competency of healthcare providers for disability inclusive Sexual and Reproductive Health Services in rural Nepal.”

We are thankful to reviewers for their encouraging words and praises related to manuscript and also identifying areas of improvement. A revised version (clean and track changes) reflecting the point-by-point response to the reviewers’ comments have been submitted for your consideration. We believe that we have addressed all the comments and that the manuscript has significantly improved and ready to get published.

Should you require any additional information, please do not hesitate to let me know. Thank you very much for your time and kind considerations. 

Your sincerely,

Pabitra Neupane

neupanepavitra@gmail.com

RESPONSE TO THE REVIEWERS

Journal Requirements:

Please review your reference list to ensure that it is complete and correct. If you have cited papers that have been retracted, please include the rationale for doing so in the manuscript text or remove these references and replace them with relevant current references. Any changes to the reference list should be mentioned in the rebuttal letter that accompanies your revised manuscript. If you need to cite a retracted article, indicate the article’s retracted status in the References list and also include a citation and full reference for the retraction notice.

Response: The reference list has been checked and the reference is updated accordingly.

Answer:

Reviewer’s Comment

Reviewer #3: The authors have successfully addressed the feedback provided by the previous reviewers, demonstrating commendable engagement with critical suggestions. Upon reviewing this revised submission, I recommend it for publication. This study is particularly notable for its focus on a critically underexplored area disability-inclusive sexual and reproductive health services in rural Nepal. The authors' examination of healthcare providers' competencies in delivering these services is highly relevant, especially given the persisting barriers faced by women with disabilities, including knowledge gaps in disability management, biased perceptions, and inadequate communication skills.

The use of a qualitative, thematic research design, supported by key informant interviews and focus group discussions, has allowed for an in-depth exploration of these complex issues. The findings, which highlight significant gaps in procedural skills and the need for disability-inclusive training and infrastructure, provide valuable insights for improving health systems in underserved regions.

Considering that the authors are university students, the depth and rigor of their work are commendable. They have contributed meaningfully to an area of research that has clear implications for both policy and practice in the context of rural health services in Nepal. Therefore, this manuscript contributes to the field and should be published.

Response: Thank you for your and encouraging feedback. We greatly appreciate your recognition of the importance of this research and are grateful for the opportunity to contribute to the underexplored area of disability-inclusive sexual and reproductive health services. We look forward to seeing our work published and hope it informs future policies and practices in rural health services.

Reviewer #4: Abstract

1. Result (Line 38-45): This sentence is long and hard to follow. It's important to clarify this for the reader to easily grasp the meaning.

Response: Thank you so much for your comment. The comment has been addressed in the manuscript in the line number 37-45. We have broken the single sentence into multiple for readability and clarity. 

2. Conclusion (Line 46-50): Instead of writing one long sentence (which spans four lines), consider breaking it into simpler sentences to improve clarity.

Response: Thank you so much for your feedback. The comment has been addressed in the manuscript from line number 48-54. We have broken the single sentence into multiple simpler sentences for readability. 

3. Introduction

• Line 90: There are two periods (dots).

Response: The comment has been addressed in the manuscript line number: 92. We have removed the extra dot.

4. Method (Lines 145-150):

Participants and Sampling:

• The criteria for inclusion/exclusion need more details. Do you include both males and females? What is the age range? What languages do participants speak (Nepali or English)?

Response: Thank you so much for your comment. The additional information about the participants and inclusion has been added in the manuscript, line number: 132-135, 148-151

5. Results

• Line 176: Double-check the spacing between words.

Response: Thank you for the comment, we have double checked the spacing between words and there is no any error.

• Lines 208-210: This sentence is somewhat unclear. Are healthcare providers aware of advocacy?

Response: Thank you so much for the comment. This comment has been addressed in manuscript, line number 212

6. Discussion:

• “Professionalism and Behavior of Health Care Providers”: This section presents clear points and is easier to follow compared to the previous section, “Disability-Inclusive Health System,” which I have already commented on for revision.

Response: Thank you for your positive remarks on this. We have tried to present the findings in the sections with clarity. 

7. Line 428: There is an extra period (dot).

Response: The comment has been addressed in the manuscript line number: 433. We have removed extra dot.

We have thoroughly reviewed the manuscript for language edit.

---

## [Editor Report · Decision Letter 2]

29 Sep 2024

Understanding challenges and enhancing the competency of healthcare providers for disability inclusive Sexual and Reproductive Health Services in rural Nepal

PONE-D-24-16726R2

Dear Dr. Pabitra Neupane,

We’re pleased to inform you that your manuscript has been judged scientifically suitable for publication and will be formally accepted for publication once it meets all outstanding technical requirements.

Kind regards,

Yitagesu Habtu Aweke, Ph.D

Academic Editor

PLOS ONE

---

## [Editor Report · Acceptance letter]

4 Oct 2024

PONE-D-24-16726R2 

PLOS ONE

Dear Dr. Neupane, 

I'm pleased to inform you that your manuscript has been deemed suitable for publication in PLOS ONE. Congratulations! Your manuscript is now being handed over to our production team.

Kind regards, 

on behalf of

PhD Candidate Yitagesu Habtu Aweke 

Academic Editor

PLOS ONE